# Assessing Coastal Road Flood Risk in Arctic Alaska, a Case Study from Hooper Bay

Anna Christina Miller and Thomas Michael Ravens *

Department of Civil Engineering, University of Alaska, Anchorage, AK 99508, USA; amiller17@alaska.edu
* Correspondence: tmravens@alaska.edu

**Abstract:** Rising waters and land subsidence are increasing relative sea levels in western and northern Alaska, forcing communities to relocate or armor in place. To appropriately plan and make equitable decisions, there is a need to forecast the risk of flood exposure in coastal Alaskan communities and to evaluate methods to mitigate that risk. This paper conducts use-inspired science to evaluate the current and future flood exposure of roads in Hooper Bay, Alaska, proposes a unit cost of flood exposure to estimate the cost of flooding, and compares various mitigation efforts including elevating roads and building dikes. Nine historic storms and their associated flood depths were subject to return-period analysis and modeled for several sea level rise scenarios. Based on the simulated road flood exposure (km hours/storm), and the storm-return period, an annual flood exposure (km hours/year) was computed. Then, the unit cost of flood exposure (USD/km hours) was determined as the ratio of the cost of flood mitigation (USD/year) to the annual flood exposure mitigated by the project. The analysis found that the unit cost of flood exposure, in conjunction with flood exposure calculations, does provide an approximate flood risk calculation, though a unitized cost of flood exposure needs to be divided into lump sum costs and materials costs. The analysis also found that dikes may be a more cost-effective alternative than road elevation. The flood risk calculation, based on the unit cost of flood exposure, could be made for all of the communities in a given region to identify those communities that face a high flood risk. Furthermore, if one divides the unit cost of flood exposure by the population, one obtains a cost/benefit ratio that potentially could be used to prioritize flood mitigation work.

**Keywords:** Alaska coastal flooding; Alaska flood risk estimation and mitigation

## 1. Introduction

Rising sea levels have created a critical new question for policymakers: assuming limited federal funding, how do we choose which communities to protect? Climate change in the form of sea level rise (SLR) and land subsidence is increasing relative sea levels in western and northern Alaska, and there is little consensus on how best to respond [1]. A General Accounting Office study found that 184 (86%) of Alaska Native villages were already impacted by flooding and erosion to some extent, and that four (Kivalina, Koyukuk, Newtok, and Shishmaref) were in imminent danger [2]. Flooding and erosion impacts are likely to worsen with sea level rise, and these communities, like many in the U.S., do not have the internal funds necessary to protect themselves. It is likely that the fate of these coastal communities will be determined largely by how federal and state agencies choose to distribute aid. Policymakers need to consider which villages will get funding, what mitigation strategies to employ, and how to provide the best quality of life and exit strategy for those whom funding does not cover. Options currently under consideration for the threatened communities are to create protective infrastructure, to relocate the communities, or to co-locate residents to existing communities.

Despite the urgency of this issue, there is no consensus on how to distribute funds or when to use different mitigation strategies. A paper by the Brookings Institute (2013)

found that in Alaska, "There is no adaptive governance framework in place to evaluate when communities and government agencies need to shift their work from protection in place to community relocation." There is a need for supporting research to provide more information on what risks these places will face, and what the potential costs may be, to help inform decision making.

Broader decision-making frameworks benefit from data at the community level. This paper explores that community-level assessment; risks of road flooding in the Hooper Bay community are evaluated and associated with their mitigation costs. This could be expanded to risks for additional infrastructure and coastal hazards and generate information to support multi-community funding decisions. Similar, community-level flood risks were estimated for the California [3]. Their work was based on the Coastal Storm Modeling System (CoSMoS), a suite of hydrodynamic models including Delft3D [4] and Xbeach [5], which computed the wave run-up and setup, and the consequent coastal flooding. Using wind and pressure forcing from global climate models, and accounting for sea level rise, CoSMoS predicts the coastal flood risk based on assumed valuations of coastal infrastructure. Lantz et al. provide storm-surge inundation risk for three coastal communities on the North Slope of Alaska, but their work over-calculates risk as it is a static calculation based on topography and surge height alone [6]. Actual coastal flooding is a dynamic hydrodynamic process, and flood waters require time to propagate across the land surface. Ravens and Allen used a hydrodynamic model (Delft3D) to compute coastal inundation on the Yukon-Kuskokwim Delta for historic storms, finding that flood waters required several hours to inundate the delta plain [7]. Assuming flooding of all coastal locations with elevations less than the peak surge heights leads to an overestimate of the flood risk.

The 2019 Intergovernmental Panel on Climate Change report indicates that the global mean sea level (GMSL) will rise 0.24–0.32 m by 2050, relative to levels in 2000, with relatively low uncertainty. Uncertainty increases with longer term projections from uncertainty of future emissions—medium confidence estimates range between 0.43–0.84 m by 2100. Each of these estimates uses the averages of results from two carbon emission scenarios— Representative Concentration Pathway (RCP)2.6 and RCP8.5—to define the range. The full range for RCP8.5—the scenario in which there is no great effort to reduce carbon emissions—is 0.61–1.10 m in 2100, with a possible GMSL of 2 m.

In Alaska, glacial isostatic adjustment from the last ice age, as well as recent glacial retreat, has created uplift, especially in south-central Alaska. A NOAA technical report found that relative sea level (RSL) rise along much of the Alaskan coast will be less than the predicted GMSL, primarily because of this uplift, in low and intermediate sea level rise scenarios (up to 1 m GMSL by 2100) [8]. However, on the Yukon-Kuskokwim Delta specifically, the report found RSL to be nearly equivalent to GMSL, or higher than GMSL in high and extreme scenarios.

The Yukon-Kuskokwim Delta (YKD) has low elevation and experiences frequent storms, making it particularly vulnerable to storm-surge flooding and climate change impacts. Prior to the mid-1900s mandate that Alaska Native children attend federally regulated schools, Alaska Native communities migrated between the coasts and central Alaska to follow fish and game [9]; following the mandate, permanent coastal habitations were created, sometimes in locations only intended as summer camps. Besides sea level rise, these coastal communities face climatic threats from the effects of melting permafrost, salinization, erosion, and changing ecosystems impacting a subsistence-based lifestyle. Terenzi et al. found that flooding appears to be more frequent on the YKD than other coastal areas of Arctic Alaska and Canada, probably because of the relatively low elevation (about 2 m relative to mean sea level) and the significant tidal range (2.7 m) [10].

Hooper Bay is a city and Native Village of 1375 people [11] on the Yukon-Kuskokwim (YK) Delta that experiences annual flooding from storm surges (Figure 1). The YK Delta lies on the Bering Sea, and this sea marks the southern portion of the U.S. Arctic [12]. The community is divided into the northeast "Old Town" (61°31′52″ N, 166°5′45″ W) and southwest "New Town" (61°31′40″ N, 166°6′32″ W) connected to each other and to the

airstrip by Airport Road. In 2017, the town finished a construction project that fortified the airstrip against coastal erosion and elevated the airport access road between New Town and the airstrip, improving airport accessibility during storms. The as-built plans for this project and the compilation of bids provided critical information for this study.

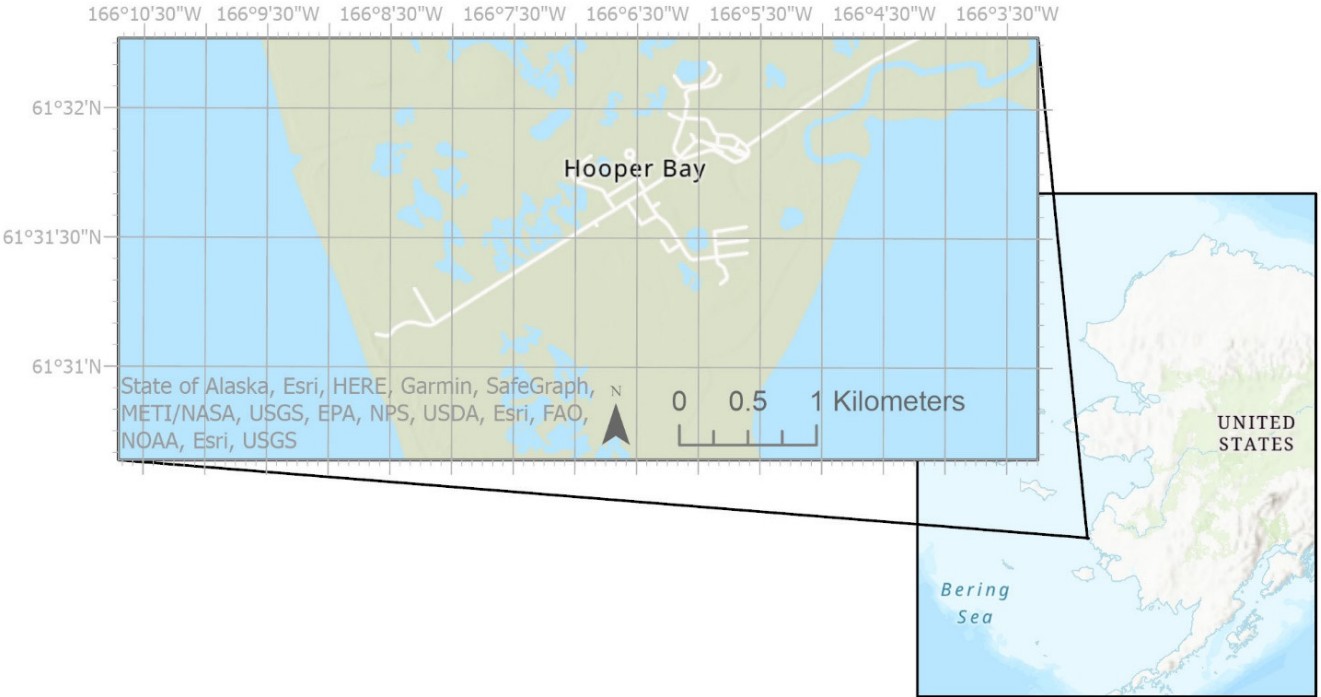

**Figure 1.** Location and Map of Hooper Bay.

Aside from elevating roads, communities may take any number of steps to provide flood protection. Flood evaluations for Shishmaref [13] compare community relocation to a new site to co-location with an existing community or creating protective measures at the original site. Relocation might take place all at once, as Newtok is doing in its move to Mertarvik, or in a staged retreat, when the community plans new infrastructure on nearby high ground while abandoning or demolishing threatened structures in a slow migration. Other options for site protection include elevating homes, creating dikes, or bolstering natural protective features at the site.

Government agencies, including the Army Corps of Engineers, have been aware of the climactic threats to coastal Alaska for some time. A 2003 Government Accountability Office study identified all the Alaska Native villages experiencing flooding and erosion and noted problems with existing relief structures [2]. Many villages do not qualify for assistance because they do not meet cost/benefit requirements (existing infrastructure has less total value than the proposed protection measures) or cost-share requirements, where the village must fund 25–50% of the project costs. Government agencies do not consistently coordinate their efforts with community plans, and money is spent on infrastructure in communities planning to relocate. Additionally, there is a lack of gauging stations and flood extent data in most villages, so that it is difficult to rank and assess communities outside of those with the direst need. The Federal Emergency Management Agency (FEMA) publishes flood maps for communities, but only if they opt into the National Flood Insurance Program (they do not have a map for Hooper Bay).

A 2013 paper from the Brookings Institute reviewed the government response to the threatened villages and identified key challenges [9]. A stand-out note is that there is currently no structure to decide when communities need to relocate vs. armor in place, and therefore protection infrastructure is being constructed in communities that may only last

a few years before the strategy shifts to relocation. There is a need for decision-making frameworks to assist in evaluating the costs of climate change.

## 2. Research Goals and Objectives

This research demonstrates a method of estimating road flood risk in coastal Alaskan communities using the community of Hooper Bay, Alaska as a case study. In the method, we first use available flood-mitigation project data and flood-exposure calculations to determine unit cost of flood exposure (USD/km hour). Second, we use high-resolution flood model calculations to determine the annual flood exposure (km hours/y), accounting for sea level rise. Finally, we produce an estimate of the flood risk of the community as the product of the unit cost of flood exposure and the flood exposure (USD/y). The method could be used generally throughout coastal Alaska to obtain an estimate of the current and future flood risk faced by communities. In addition, the research explores various flood mitigation actions in Hooper Bay and estimates their cost.

This study conducts "use-inspired research" [14] and it has the following goals: (1) to develop and assess a simple way of estimating the cost of roadway flooding using a unit cost of flood exposure approach, and (2) to estimate the cost of mitigating road flooding in Hooper Bay based on three alternative approaches. These goals will be achieved with the following four objectives. First, we determine the flood exposure of Hooper Bay roads under different sea-level-rise scenarios. This objective is accomplished by modeling the principal historic storms, by calculating the return period of storm flooding, and by determining the annual flood exposure (km hour/y). Second, we approximate the unit cost of road flood exposure (USD/km hour) based on the cost of the Hooper Bay road elevation project and on the flood exposure mitigated by the project. Third, we estimate the flood risk of Hooper Bay roads by taking the product of the unit cost of flooding and the flood exposure. The flood risk calculation we employ resembles the one provided by Tariq et al. [15]: risk (USD/y) = probability ($y^{-1}$) x consequences (USD). However, given the model-based flood exposure available to us, we chose to directly tie probability and consequences to flood exposure. Fourth, we estimate the total community road flooding protection costs for 2020–2050 for three possible mitigation approaches. The fourth objective allows us to assess the efficacy of the flood-rise estimate produced under the third objective.

## 3. Materials and Methods

### 3.1. Objective 1: Determine the Road Flood Exposure of Hooper Bay in Different SLR Scenarios

To determine the road flood exposure in Hooper Bay, nine historic storms were modeled in Delft 3D. The models were re-run with simulated sea level rise. The maximum water level from the baseline conditions was used with extreme value analysis theory to determine the annual water level probabilities and flood-return period. Finally, the annual flood exposure was calculated for the community with units of km hours/year.

#### 3.1.1. Storm Modeling

Nine historic storms were modeled on a 200 km × 150 km grid using Delft3D software (Figure 2). The ocean boundary of the Delft3D model was provided by a USACE ADvanced CIRCulation (ADCIRC) model [16,17], and supplemented with tidal fluctuations from tidal analysis of local water level data. The majority of the model grid had 170 m resolution, but the model included a 20 m resolution grid for the City of Hooper Bay (Figure 2). The same nine storms were re-modeled with sea level rise simulated by adding 0.3 m to the boundary conditions—the assumed conditions for 2050. The storms were from October 1992, October 1995, October 1996, November 1996, October 2004, September 2005, two from October 2006, and from November 2011. A subset of the storms was also modeled with 60, 90, and 120 cm of sea level rise (October 1995, September 2005, and November 2011).

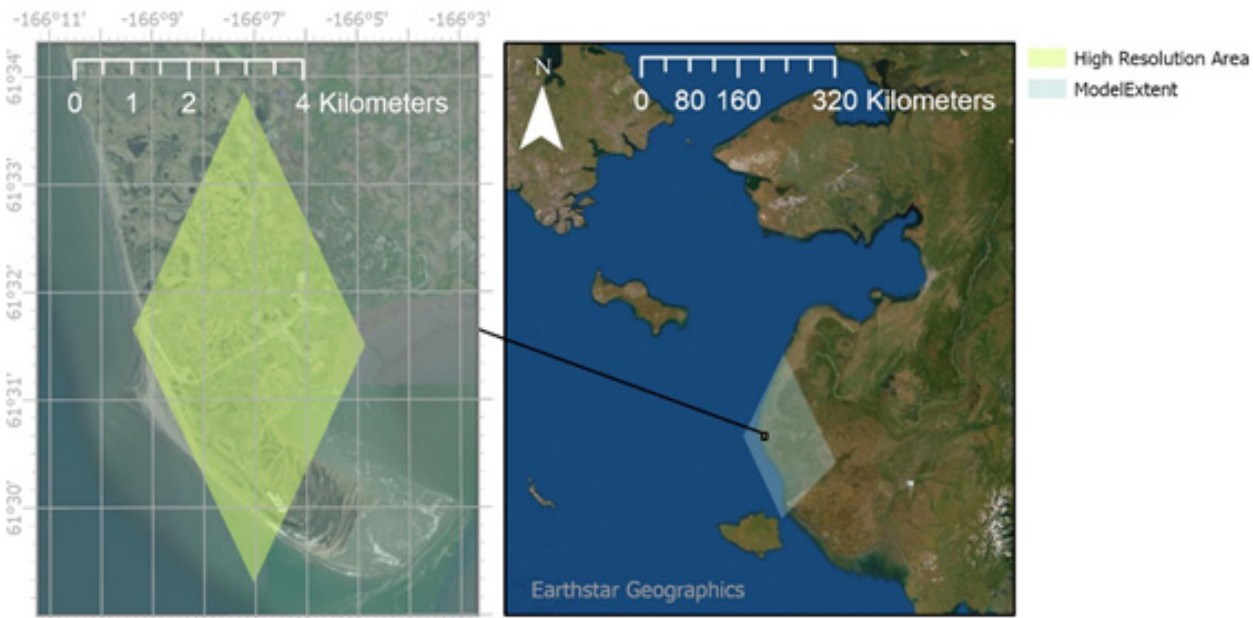

**Figure 2.** Delft3D model extent (**right**) and high-resolution model domain for Hooper Bay (**left**).

Work was done to verify that the models produced reasonably accurate water levels for the simulated storms. While there is little data for model validation, the Division of Geological and Geophysical Surveys (DGGS) has compiled a repository of storm images, created a report collecting anecdotes about storm height and severity, and made a map of approximate water levels for different storms [18]. To determine if the model is producing reasonably accurate results, model storm simulations were compared to these assessments. The anecdotal evidence and map appear to indicate that the model outputs are reasonably correct (water elevations within 0.5 m of observations). Of note, the DGGS investigation found that the 2005 storm led to flooding of the fuel tank storage area, and that this was on an edge of the flooding. Modeling the 2005 storm does show that at the peak of the storm there would be some flooding in the fuel tank area, and that this is a flood edge.

### 3.1.2. Return Period Calculation

Extreme value analysis was conducted following Goda's method [19] to determine the return period of the nine storms. The return period provides the likelihood that a flood of a particular extremity will occur in a given year; for example, a 100-year flood would have a 1/100 chance. Alternatively, the return period can be considered the recurrence interval, which is the expected average time between storms of that magnitude. It is expected that at least one storm of that magnitude occurs within the return period. There are several notes that should be mentioned in this application. First, extreme value analysis is a statistical analysis that is dependent upon sample size; the smaller the sample size, the larger the possible variability and resulting confidence interval. With limited flood data, we are necessarily limited to a small sample size and will use this as a best first attempt at flood-return period estimation. This limited dataset uses the annual maxima and peak over threshold methods (which have identical techniques). There is also difficulty in shifting the extreme value analysis from waves (Goda's method) to flooding. This change has an unknown interaction with constants determined by Goda for extreme value analysis of waves using a Monte Carlo analysis; it likely has little effect on the distribution and return-value calculation but does impact the calculation of the confidence intervals of the distribution.

Previous studies have analyzed the return periods of storms in Hooper Bay based upon the storm surge height of the storms [17]. For this work, it was important to evaluate the return period based upon flood impact, as storm-surge height (at the coast) does not

always indicate the severity of flooding. Flood extremity was estimated based on the maximum water level reached in the modeling area. These values ranged between 3.06 and 4.17 m above Mean Sea Level (MSL).

The first step of the extreme value analysis is to calculate the expected non-exceedance probability of each data variate using the plotting position formula, assuming a data distribution with the data ordered from largest to smallest. The unbiased plotting position formula recommended in Goda's approach is as follows:

$$F_{(m)} = 1 - \frac{m - \alpha}{N_T + \beta} \tag{1}$$

where $m$ is the rank of the data (1, 2, 3 … $n$) with data ordered in descending size, and $N_T$ is the total number of storm events during the period of observation. For this study, it is assumed that we have analyzed all major storms between 1992 and 2011: $N_T$ is the number of storms (9). Constants $\alpha$ and $\beta$ are defined by the assumed distribution, according to Table 1 below. These constants require definition of the shape parameter k for the Fréchet and Weibull distributions. For this analysis, k values of 2.5, 3.33, 5, and 10 were tested for the Fréchet distribution, and of 0.75, 1, 1.4, and 2 were tested for the Weibull distribution, with the fixed values recommended by Goda.

**Table 1.** Coefficient values and reduced variate for extreme value distributions.

| Distribution | $\alpha$ | $\beta$ | Reduced Variate Equation |
|---|---|---|---|
| Gumbel | 0.44 | 0.12 | $y_{(m)} = -\ln[-\ln(F_{(m)})]$ |
| Fréchet | $0.44 + 0.52/k$ | $0.12 - 0.11/k$ | $y_{(m)} = k[(-\ln(F_{(m)}))^{-1/k} - 1]$ |
| Weibull | $0.2 + 0.27/\sqrt{k}$ | $0.2 + 0.23/\sqrt{k}$ | $y_{(m)} = [-\ln(1 - F_{(m)})]^{1/k}$ |

From the expected non-exceedance values $F_{(m)}$, we calculate the reduced variate (the flooded area expected to have the calculated non-exceedance value for each distribution). The correlation between the peak water level in a given storm ($x_{(m)}$) and the reduced variate values ($y_{(m)}$) is calculated, with the correlation closest to 1 indicating the best-fitting distribution.

In the analysis, the Weibull distribution with $k = 2$ was found to best fit the data. The scale parameter A and the location parameter B were determined using linear regression, plotting the original $x_{(m)}$ data vs. the reduced variate $y_{(m)}$ data for the Weibull distribution and creating a trend line with Equation (2).

$$x_{(m)} = B + Ay_{(m)} \tag{2}$$

Note that because of the nomenclature this appears slightly different from the typical linear equation y = mx + b. The plot comparing the reduced variate to the flooded data is shown in Figure 3 below. The $R^2$ is 0.92.

With all parameters defined, the Weibull distribution is applied to the data to generate the cumulative distribution function. The Weibull distribution equation is as follows:

$$F(x) = 1 - exp\left[-\left(\frac{x - B}{A}\right)^k\right] \quad for\ B \leq x \leq \infty \tag{3}$$

To find the return period based on the cumulative distribution, each year is assumed to be divided into segments based on the mean rate of extreme events $\lambda$. This is the expected number of events per year (i.e., the number of events in the period of analysis divided by the years in the period of analysis). In this analysis, $\lambda$ has a value of 0.47, found by dividing the nine events by the 19 years between the first storm analyzed and the last. The

equation to calculate the return period R is Equation (4) below, providing the associated return period for each storm:

$$R = \frac{1}{\lambda[1 - F(x)]} \tag{4}$$

The confidence intervals for the return values are very large, primarily because of the small sample sized used. However, it should be noted that the actual confidence intervals are even larger; there is currently no method to determine the true parent distribution of flood exposure beyond trying several different distributions, as we have done here. If the distribution is improperly fitted, the confidence intervals are larger than calculated using the following equations.

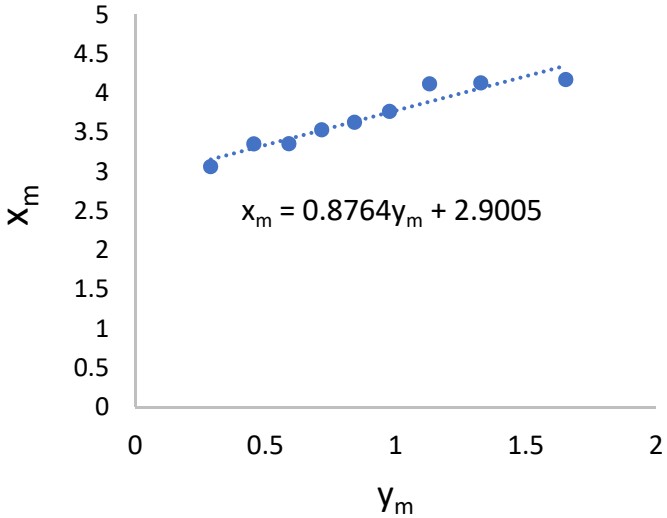

**Figure 3.** Fit of the Weibull k = 2 distribution.

From the parent distribution, a new reduced variate is calculated. For the Weibull distribution, this is given by the Equation (5), where $R$ is the return period:

$$y_R = [\ln(\lambda R)]^{-1/k} \tag{5}$$

This is used with constants determined by Goda in the Monte Carlo simulations to calculate the standard deviation for the reduced variate. As mentioned earlier in this section, it is unclear how these constants affect the confidence interval on values that are not wave datasets. However, the flood depths used have a similar magnitude as extreme waves, so it is possible that the difference has little impact. The constants for the Weibull distribution with a shape parameter of 2 are provided in Table 2:

**Table 2.** Constants for the Weibull distribution, $k$ = 2.

| Constant | Value or Equation |
|---|---|
| $a_1$ | 2.24 |
| $a_2$ | 11.4 |
| $\kappa$ | 1.34 |
| $c$ | 0.5 |
| $\alpha$ | 0.54 |
| $\nu$ | 1 |
| $a$ | $a_1 \exp\left[a_2 N^{-1.3} + \kappa(-\ln \nu)^2\right]$ |

The standard deviation for the reduced variate is given in Equation (6):

$$\sigma_x = \frac{[1.0 + a(y_R - c + \alpha \ln \nu)^2]^{1/2}}{\sqrt{N}} \tag{6}$$

The standard deviation of the return value is then the dot product of the standard deviation of the reduced variate with itself, as derived by Goda.

$$\sigma(\hat{x}_R) = \sigma_x \bullet \sigma_x \tag{7}$$

This is applied to each return value to calculate the confidence intervals around the distribution fit line.

For sea-level-rise scenarios, it is assumed that the return period remains constant—that is, the storm modeled with sea-level-rise conditions would have the same return period as the storm modeled with baseline conditions. This may be inaccurate, as climate change is affecting the character and frequency of storms.

### 3.1.3. Flood Mapping

Water level data from all the storms were analyzed in MATLAB to produce point data to import into Geographic Information System (GIS) format. The MATLAB code compiled the maximum water level that occurred at each Delft grid location for all the storms, and each sea-level-rise scenario. This point data was imported into GIS with an Alaska Albers equal area conic projection and used to create a raster with 20 m bin sizes, depicting the maximum depth of flooding at all locations within the community. These maps are an important output of the study, as they demonstrate the flood depth in different sea-level-rise scenarios and allow visualization of the spatial extent of flooding. The maps were used to find the roads vulnerable to flooding now and in 2050, identifying key locations for observation points in the model data. Figure 4 shows the locations of the observation points. From the model, flood depth and time data were obtained at each observation point for all the storms in baseline and 2050 conditions (30 cm SLR).

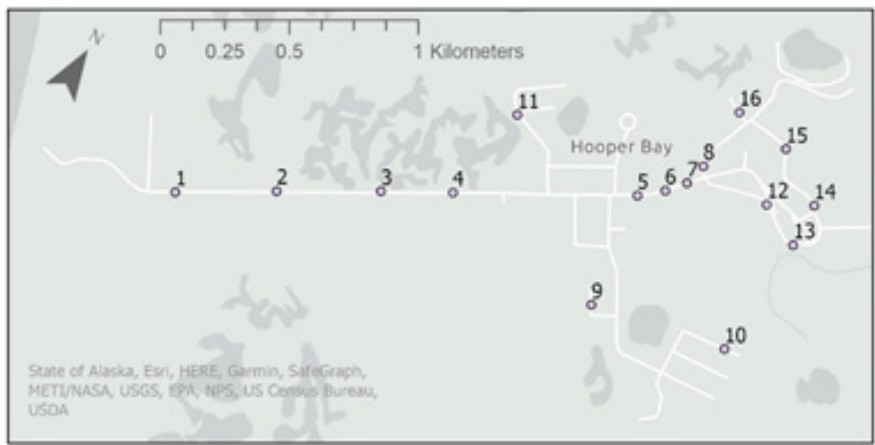

**Figure 4.** Model observation points.

The storms are modeled using road elevations prior to the elevation project, allowing us to measure flood exposure and determine the mitigation impact of the project. Four observation points (1–4) were placed along the airport access road, three (5–7) were placed on the Old Town/New Town road, three (9–11) are on vulnerable New Town roads, and six (8, 12–16) are on vulnerable Old Town roads. Of particular interest are observation points 8, which is next to the fuel storage area, and 16, which is close to the water plant.

For roads, the critical impacts of flooding are damage (wash-out) and closure time. It is assumed in this case study that the flooding does not cause any damage, so that the impact is solely due to the hours the community is unable to use the road. In Hooper Bay, the road network is comparatively simple; there is one main road that runs through the entire community. If any part of the road is impassable, it prevents community members from reaching critical resources (the water plant, power plant, and airport); therefore, the entire road can be treated with the same minimum allowed flooding. During a large flood event, this mainland community becomes disconnected islands, as shown below in Figure 5.

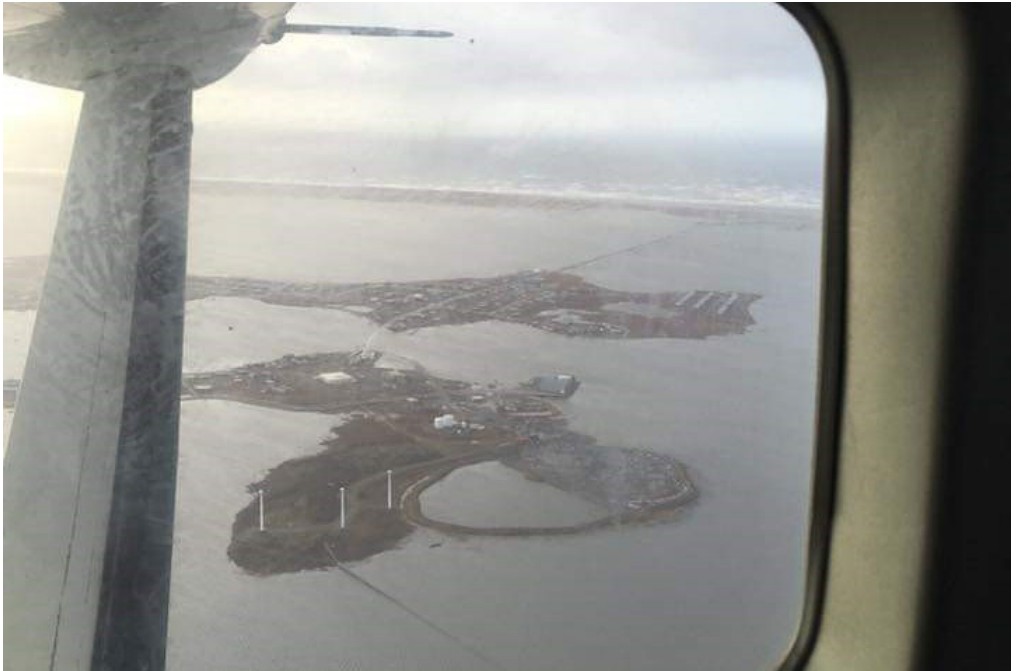

**Figure 5.** Hooper Bay flooding (photo by William Naneng, 2016, used with permission).

In other communities, analyzing road systems can be very complicated. A network of roads may allow multiple routes to resources, and flooding of one road may be inconsequential for the community at large. Different roads or networks of roads may have different criticalities; for example, roads that lead to a park vs. roads that lead to the water facility. In another layer of complexity, a resource may technically be accessible, but access becomes more difficult with multiple flooded roads. In these other communities, assessing the impact of road flooding may require more in-depth criticality analysis.

In this case study, with all points on the road having equal criticality, flood exposure will be measured in kilometer-hours. The hours of flooding will be determined by the time the road is impassable, while kilometers measure the length of road affected. With this kilometer-hours metric, a storm that floods the entire community for an hour is rightly shown to be more severe than a storm that floods a small portion of the community for the same amount of time. The general recommended maximum flood depth for driving is 10 cm, so we define any flooding greater than 10 cm to be impassable. Multiplying the hours of flooding at each point by the represented kilometers of road (shown in Table 3), we find the total kilometer-hours of impassable flooding for each storm. The represented kilometers for each observation point were estimated by dividing the total length of the road affected by flooding (estimated using GIS distance measurement tools) into sections around each observation point.

**Table 3.** Observation point information.

| Name | Latitude (Y, deg) | Longitude (X, deg) | Estimated Road Length (m) |
|---|---|---|---|
| Obs 1 | 61.51978 | −166.13460 | 375 |
| Obs 2 | 61.5218901 | −166.1278690 | 375 |
| Obs 3 | 61.523828 | −166.1215510 | 375 |
| Obs 4 | 61.5251258 | −166.1171688 | 375 |
| Obs 5 | 61.5284810 | −166.1059030 | 10 |
| Obs 6 | 61.5291370 | −166.104405 | 100 |
| Obs 7 | 61.5297640 | −166.103379 | 100 |
| Obs 8 | 61.530556 | 166.103056 | 215 |
| Obs 9 | 61.524471 | −166.104426 | 140 |
| Obs 10 | 61.525671 | −166.094672 | 130 |
| Obs 11 | 61.528566 | −166.116316 | 100 |
| Obs 12 | 61.530628 | −166.097743 | 330 |
| Obs 13 | 61.529957 | −166.094571 | 220 |
| Obs 14 | 61.531475 | −166.094842 | 390 |
| Obs 15 | 61.532585 | −166.098755 | 220 |
| Obs 16 | 61.532774 | −166.103008 | 100 |

### 3.1.4. Annual Flood Exposure

The return period (*R*) is used with the kilometer-hours of inundation value (*H*) for each storm to find an annual weighted average for the kilometer-hours of inundation, here referred to as annual flood exposure (*AFE*); the kilometer-hours of flooding are anticipated on an annual basis. The following equation is used, where *N* is the number of storms and i is an index value:

$$AFE = \frac{\sum_{i=1}^{N}(H_i/R_i)}{\sum_{i=1}^{N}(1/R_i)} \tag{8}$$

This annual expected value is calculated for each sea-level-rise scenario to calculate the flood exposure in different sea-level-rise conditions.

### 3.2. Objective 2: Approximate the Unit Cost of Road Flood Exposure (USD/km hr) from the Hooper Bay Airport Improvements Project Access Road Elevation from 2020–2050

The unit cost of flood exposure for the access road is the cost of the project divided by the years of protection it provides, normalized to the average annual flood exposure it prevents. The equation used is:

$$(Cost\ of\ the\ project/years\ of\ effective\ protection)\ /\ (km-h/year\ of\ flood\ exposure) = \$/km-h \tag{9}$$

This is a unitized cost of flood exposure mitigation through roadway elevation, assuming that the mitigation is effective. Obtaining this number requires several steps. First, the cost of the elevation project was estimated from the compilation of bids. Next, the number of years that the road elevation will be effective was determined. Finally, the annual flood exposure for the access road prior to elevation (observation points 1–4) was found for both baseline conditions and the final year that the elevation effectively prevents flooding, and the two values were averaged together; this is an estimate of the average flood exposure that the roadway elevation prevented. The access road's annual flood exposure was found using the method described above, but the cost estimation and determination of the years of effectiveness merit further explanation.

Cost estimation for construction in the Yukon-Kuskokwim Delta is challenging. Its remote location means that cost estimates used for construction in the rest of the United States, or even other parts of Alaska, are likely to be inappropriate. Costs were estimated based on the average cost of line items from the compilation of bids for the Hooper Bay airport renovation project, which included the estimates of three engineering contractors, as well as an estimate from the state. The compilation includes costs associated with modifications made to the airport and runway, as well as to the airport access road; the

line items with high likelihood of association with the access road elevation project were separated out, and average unit price identified. These unit prices were used with quantity estimates to complete a rough construction cost estimate for the access road and was adjusted for inflation from 2015 to 2020 U.S. dollars using the U.S. Bureau of Labor Statistics Inflation Calculator [20].

To calculate the years of effectiveness for the access road elevation project, flood data for observation points 1–4 were modified to reflect the new elevation of the road. The elevation added at observation points 1–4 was determined by comparing the as-built plans from Hooper Bay to the latitude and longitude coordinates of the observation points. This elevation was subtracted from the water depths at observations 1–4, to approximate the flood depth on the elevated roadway. Annual flood exposure was calculated with these elevated road conditions, and it was found that the access road began to flood just at the 30 cm SLR conditions; it is estimated that the roadway will prevent flooding through 2050.

*3.3. Objective 3: Estimate the Total Community Road Flooding Protection Costs for 2020–2050 for Three Possible Mitigation Approaches and Compare Them*

To answer the third objective, cost estimation was applied to three possible mitigation approaches for 2050; elevating the roadways affected by flooding, using a combination of road elevation, and building dikes next to the roads, and elevating some roads and constructing a larger dike that would protect all the communities and possibly reclaim flooded land. For perspective, these measures are also compared to the scenarios of "doing nothing" and relocating the community.

3.3.1. Rough Design of Mitigation Measures

Without a detailed design, it is difficult to know how high road and elevations need to be to prevent run-up. For the purposes of this rough estimate, it is assumed that any elevation higher than the modeled water level at the location would not flood. Elevations for the roads and dikes were approximated based on the estimated maximum water depth in the 30 cm SLR simulation from the flooding maps. For the roads, elevations were estimated from the observation points (listed in Table 4 below). Elevations for the dikes were estimated from all the modeled points that intersect the dike path and were classified into depth categories of 0.25 m between 0 and 2.75 m (conservatively: for example, if a point along the dike experiences a maximum flood level of 0.6, the required dike elevation at that point is estimated to be 0.75 m). Note that observation 5 did not experience flooding in any of the storm scenarios and is therefore not elevated.

**Table 4.** Simulated elevation added to roads.

| Observation Point | Elevation Added (m) |
|:---:|:---:|
| 6 | 0.5 |
| 7 | 1.1 |
| 8 | 1.1 |
| 9 | 0.3 |
| 10 | 0.7 |
| 11 | 0.5 |
| 12 | 0.9 |
| 13 | 2.7 |
| 14 | 1.1 |
| 15 | 0.9 |
| 16 | 0.9 |

Both dike elevation measures include elevating some roadways; only a few places in New Town experience flooding and building a dike to protect all the roadways is more expensive than just elevating the few segments that experience flooding. The estimated design for a road-elevation/dike combination mitigation is shown in Figure 6, while the

modeled points that intersect the dike are shown in Figure 7. The length of the dike is 2364 m.

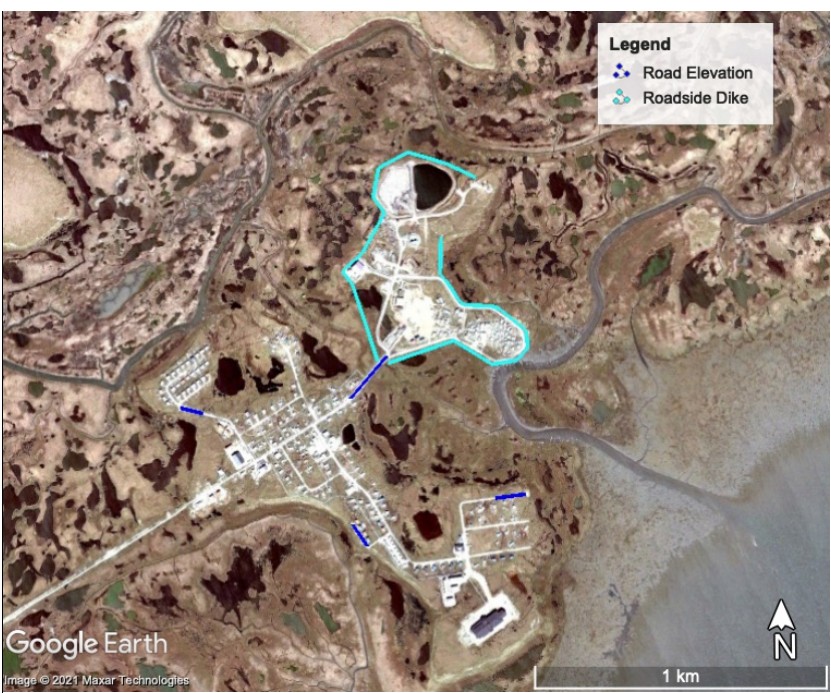

**Figure 6.** Planned roadside dike mitigation.

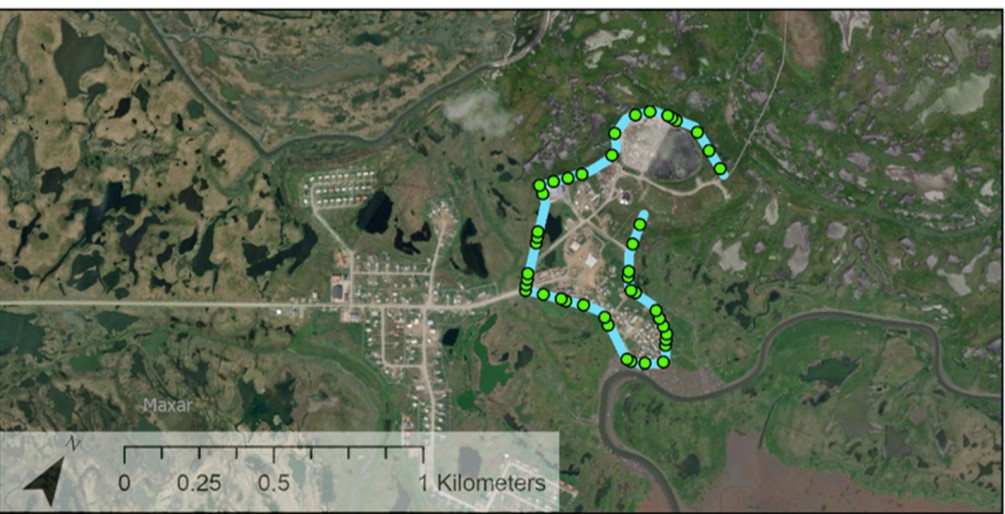

**Figure 7.** Model points used to estimate the roadside dike height.

The large dike design is shown in Figure 8, and the intersecting points are shown in Figure 9 below. The length of the dike is 3032 m.

It is assumed that the roads are cemented, and that the dikes are mounds of earth. Both are assumed to have armoring on one side. Volumes of the added material are approximated as a trapezoid, with the bottom width twice that of the top.

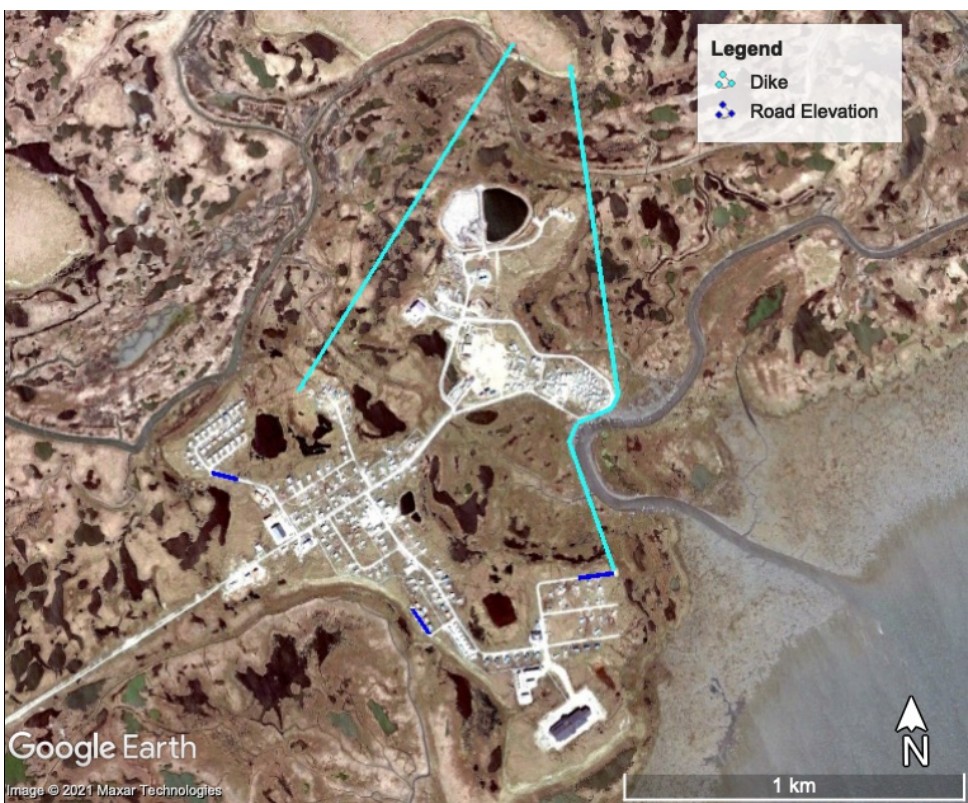

**Figure 8.** Planned large dike mitigation.

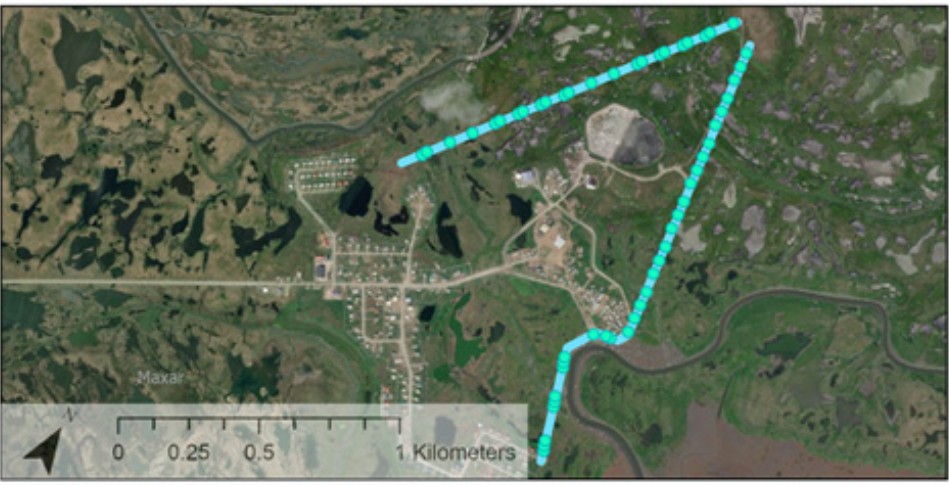

**Figure 9.** Model points used to estimate the large dike height.

Observing the peak flood images and the storm animations, most flooding appears to approach from the southeast. The exceptions are the storms from October 1992, October 2004, and September 2005, which also had flooding from the east and west. Two additional models were created for the September 2005 storm using the Delft 3D thin dam feature, one modeling the elevation of the airport access road, and one modeling a dike protecting Old Town to the east. The first model showed that it is likely that the access road elevation prevents flood waters from crossing the road, but this does not appear to significantly impact flooding in other parts of the community. The dike that only protected Old Town to the east showed that the dike protected Old Town from the first wave of flooding in the 2005 storm (which came from the east), but provided no protection and prevented draining for the wave that approached the community from the west. Because flooding comes from

multiple directions, dikes would need to surround the community to provide adequate protection, as shown in the estimated designs.

### 3.3.2. Cost Estimation of Mitigation Measures

Costs were estimated using two different approaches. First, the total community road elevation costs were estimated using the unit cost of flood exposure calculated for the airport access road. Next, the road elevation costs and dike costs were estimated based on the compilation of bids for the airport improvement project. By estimating the costs to elevate the roadways from the compilation of bids, we can assess whether the unit cost of flood exposure calculated in the second goal provides a good estimate of total community road elevation costs. Using the compilation of bids to estimate costs for the road elevations and dikes addresses some of the difficulties and uncertainties from estimating construction costs in remote Alaska. Line items from the compilation of bids were adjusted by area, length, and volume, when necessary, to approximate the costs of the new designs. It should be noted that the COVID-19 pandemic has significantly affected material costs—even when basing costs on a project in the same community, there is high uncertainty in this estimate.

The only costs considered in this estimation are the capital costs. When considering true costs over longer timeframes, it is important to include maintenance costs. Roads develop potholes, dikes may erode, and other normal wear and tear occur that compromise the integrity of the mitigation measures. The lifetime of the road is dependent on the material, traffic, and environmental conditions, and is beyond the scope of this work to assess. These costs may have interesting political ramifications; if the community uses federal or state program assistance to construct the original protection measure, should the community or the program cover the maintenance costs? For this case study, maintenance costs are not considered in our 30-year timeframe.

For perspective, a relocation cost was also considered. In the short 30-year time frame that we are planning for in this case study, relocation is too expensive to be considered a viable option. Additionally, relocation solves more coastal threats than just flooding, like erosion and salinization, which are not considered in this work; comparing relocation directly to these measures does not demonstrate the added value of the move. However, as community planners look farther into the future, protection measures like road elevation and dikes to protect against sea level rise become more costly and measures with a larger capital cost like relocation, or creating Dutch-style polders, become more financially attractive for long-term protection and community expansion. As a point of reference, relocation costs are briefly considered here.

Estimating relocation costs, particularly without a relocation site in mind, is challenging. The USACE has estimated relocation costs for Newtok (pop. 354), Kivalina (pop. 683), and Shishmaref (pop. 576) to be USD150 million, USD270 million, and USD280 million, respectively [21,22]. Assuming that population is the driver for relocation costs, linear interpolation leads to a Hooper Bay (pop. 1375) to be USD690 million—an amount that dwarfs the flood mitigation costs under consideration here.

### 3.3.3. AFE following Mitigation Measures

It is difficult to know the actual annual flood exposure following the mitigation measures without detailed design, or new Light Detection and Ranging (LIDAR) data to determine the impact of the airport road elevation. However, it may be helpful to estimate potential impact. The same estimation technique used to find the AFE of the access road to determine its years of efficacy was applied to the other roads: the added elevation was subtracted from flood depths to estimate the new flood depth at each point.

Finding the AFE for the dikes was not attempted. It is assumed that they reduce AFE to 0 in the 30 cm SLR scenario that they were designed for, and for the baseline scenario. It is likely that they also significantly reduce AFE in the 60, 90, and 120 SLR scenarios by limiting the volume of water that passes over them, but the exact amount is unknown, and not estimated.

## 4. Results

*4.1. Objective 1 Results: Determine the Road Flood Exposure of Hooper Bay in Different Sea Level Rise Scenarios*

The primary output of this goal is the road flood exposure, but there were several steps in reaching this metric that have interesting results of their own; the flooding return-period calculations and the flood maps, which also informed goals two and three, are particularly notable.

### 4.1.1. Return Period Analysis Results

Shape parameters and $R^2$ correlation values for the return-period analysis are summarized in Table 5.

**Table 5.** Correlation values for each fitted distribution.

| Distribution | k | $R^2$ |
|:---:|:---:|:---:|
| Weibull | 2.0 | 0.92 |
| Gumbel | n/a | 0.89 |
| Weibull | 1.4 | 0.87 |
| Fréchet | 10 | 0.85 |
| Weibull | 1.0 | 0.79 |
| Fréchet | 5.0 | 0.79 |
| Fréchet | 3.33 | 0.72 |
| Weibull | 0.75 | 0.68 |
| Fréchet | 2.5 | 0.65 |

It was found that the Weibull distribution with a shape parameter of 2.0 fit the data best, with an $R^2$ value of 0.92, scale parameter A of 0.87, and location parameter B of 2.9. Applying this distribution results in the flood extent return periods summarized in Table 6. Also included is the Chapman 2009 storm ranking, based on storm surge heights between 1954 and 2004, for storms that both studies shared.

**Table 6.** Storm-return periods.

| Storm | Max Water Level (m) | Return Period (Years) | Chapman Results (Storm Surge Height) | |
|:---:|:---:|:---:|:---:|:---:|
| | | | Ranking | Return Period (Years) |
| September 2005 | 4.17 | 17.21 | | |
| November 1996 | 4.13 | 15.00 | 4 | 10 |
| October 1995 | 4.12 | 14.45 | 3 | 11 |
| October 1992 | 3.76 | 5.57 | 2 | 13 |
| October 2004 | 3.62 | 4.19 | 1 | 15 |
| November 2011 | 3.53 | 3.54 | | |
| October 2006 A | 3.35 | 2.76 | | |
| October 2006 B | 3.35 | 2.74 | | |
| October 1996 | 3.06 | 2.18 | 5 | 7 |

The difference in the two rankings may be from differences between the return period of storm-surge heights and storm-surge flooding. Community flooding, for example, is significantly affected by the direction of the storm's approach; high storm surges alone do not necessarily result in flooding. Figure 10 shows the summary of the extreme value analysis.

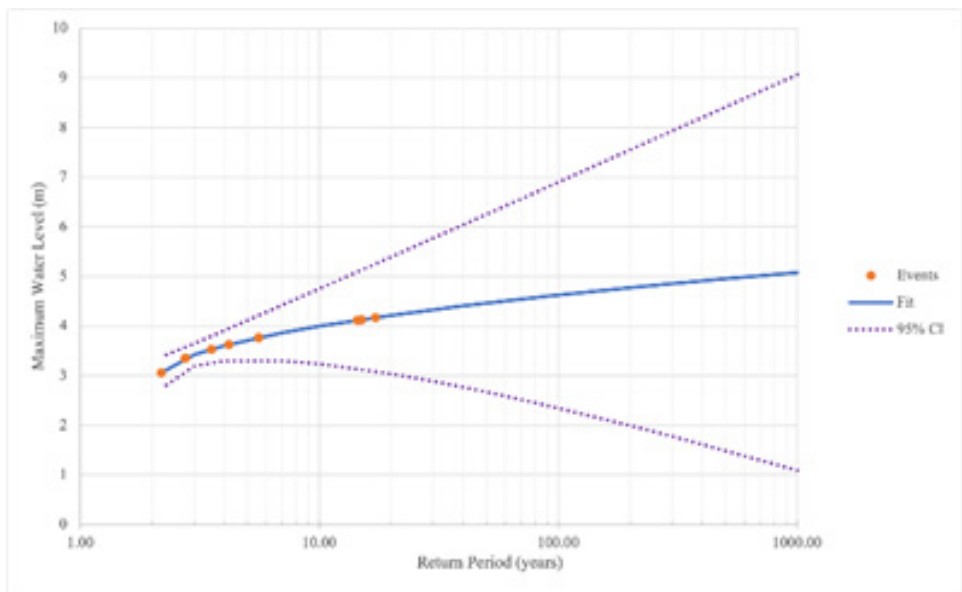

**Figure 10.** Weibull distribution with *k* = 2 applied to the nine storms.

As expected, the confidence interval for our small sample size is quite large. Additional flood modeling would improve the accuracy of this estimate. It is also evident that the fit is not entirely linear, as we would want it to be. This is examined further in the discussion in the Section 4.1.2 map of maximum flood extent and depth.

A map depicting maximum flood extent and depth for the 30 cm SLR condition (considering the nine storms simulated) is shown in Figure 11 below. An additional map was created to show the edges of the flooding from the different sea-level-rise conditions on Old Town and New Town (Figure 12).

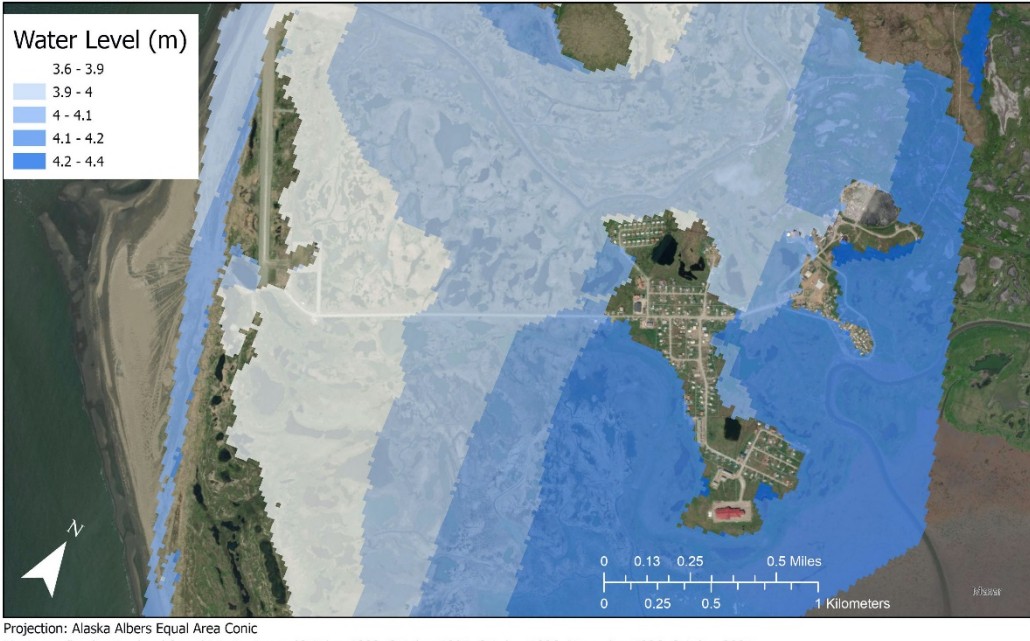

Projection: Alaska Albers Equal Area Conic
Maximum flood water level from historic storms (October 1992, October 1995, October 1996, November 1996, October 2004, September 2005, October 2006, and November 2011) modeled with 30cm of sea level rise. Map exceeds modeled area. Floods were modeled in a 20m spatial resolution grid in Delft3D software. Elevations are based on LIDAR data prior to the 2015-2017 airport improvements project, which elevated the airport access road. Vertical datum is mean sea level.

**Figure 11.** Maximum water level for the storms with a 30 cm SLR applied.

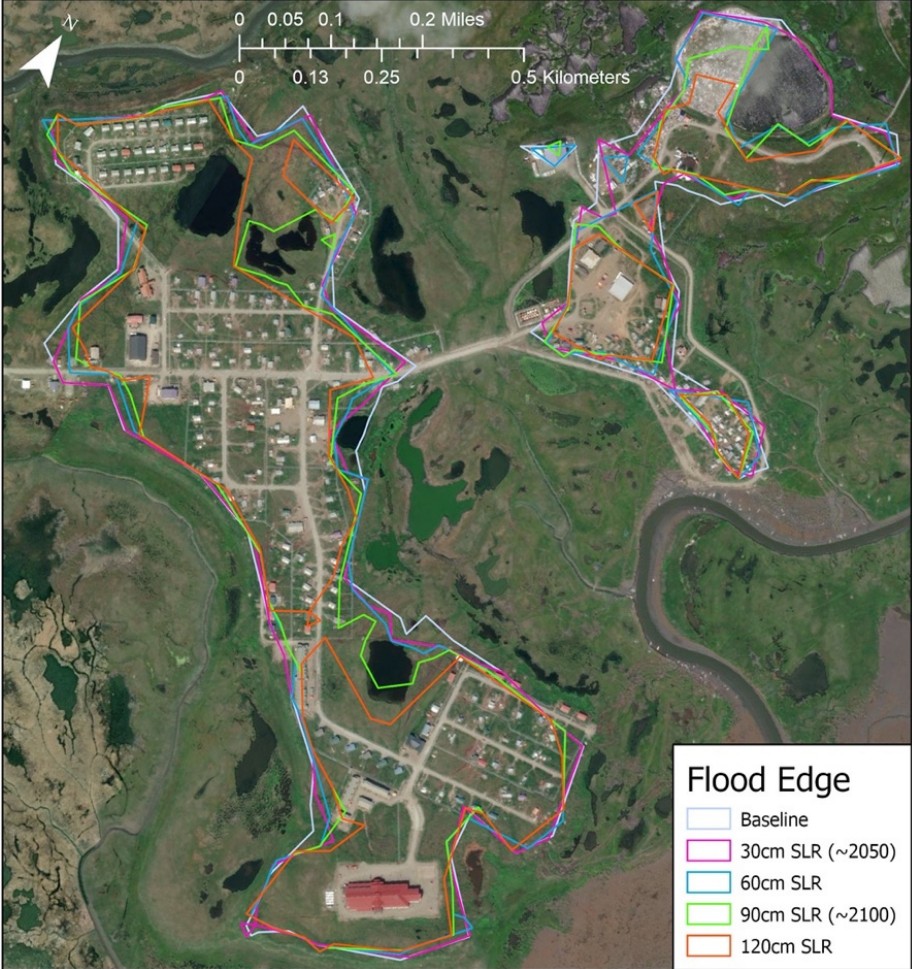

**Figure 12.** Flooding waterline in Hooper Bay.

From these maps, we can see the vulnerable locations in the community and how sea level rise will turn the community into a series of islands under flooded conditions.

### 4.1.2. Annual Flood Exposure

Annual flood exposure for each sea-level-rise scenario is summarized in Table 7 below. It shows the flood exposure before and after the access road elevation for the entire community, the access road only, and the community excluding the access road.

**Table 7.** AFE (km hours/year) for Hooper Bay.

| SLR Scenario | Baseline | 30 cm | 60 cm | 90 cm | 120 cm |
|---|---|---|---|---|---|
| Prior to Elevation | | | | | |
| All observation points | 20.6 | 32.1 | 74.3 | 98.2 | 119.5 |
| Access road only (1–4) | 6.8 | 12.3 | 35.6 | 44.7 | 51.3 |
| Access road excluded (5–16) | 13.7 | 19.8 | 38.7 | 53.5 | 68.2 |
| Post elevation | | | | | |
| All observation points | 13.8 | 20.0 | 40.0 | 57.6 | 79.0 |
| Access road only (1–4) | 0.0 | 0.2 | 1.3 | 4.1 | 6.6 |

Observation point 5 never flooded during any of the modeled storms, including those with 120 cm of simulated SLR. This point was closest to the community itself and suggests that at least parts of Old Town are sufficiently elevated to be protected well into the future.

Additionally, the October 1996 flood caused no flooding on the observation points, in either the baseline or 30 cm SLR conditions. The access road after the elevation project begins experiencing some flooding at the 30 cm SLR scenario, but very little (some flooding at observation points 1 and 2, resulting in an AFE of 0.2). If 0 flooding is acceptable, this informs the number of years that the elevation project will be effective protection (just under 30). The 0 baseline value for the access road post-elevation is rounded down from 0.04; this is why the post-elevation rounded value of 13.8 differs from the pre-elevation 13.7.

The 2005 storm was modeled with a thin dam along the access road to examine the assumption that the access road elevation did not impact flooding in other parts of the community. This assumption was shown to be true for the 2005 storm; observation points not on the access road did not experience a change in flooding.

*4.2. Objective 2 Results: Approximate the Unit Cost of Road Flood Exposure (USD/km hr) from the Hooper Bay Airport Improvements Project across Road Elevation form 2020–2050*

The unit cost of exposure came from estimating the cost of the airport access road elevation project from the compilation of bids, determining the number of years the airport road would remain above flood levels, and calculating the average annual kilometer-hours of flood exposure for the access road between 2020 and 2050 without the elevation. These values and the final unit cost are given in Table 8.

**Table 8.** Values informing the unit cost of flood exposure.

| | |
|---|---|
| Cost of the elevation project (USD 2020) | 7,998,500 |
| Years of effective protection | 30 |
| Average annual flood exposure (km h/year) | 9.57 |
| Unit cost of flood exposure (USD/km h) | 27,873 |

*4.3. Objective 3 Results: Estimate the Total Community Road Flooding Protection Costs for 2020–2050 for Three Possible Mitigation Approaches and Compare Them*

The cost estimation for each mitigation measure based on the compilation of bids, along with the estimated community AFE following the mitigation measure, is summarized in Table 9 below. Note that the calculated AFE is approximate and excludes the airport access road (which is not protected by the mitigation measures).

**Table 9.** Estimated costs and AFE excluding the airport access road.

| Measure | Cost (USD 2020) | Estimated AFE, SLR Scenarios (km h/year) | | | | |
|---|---|---|---|---|---|---|
| | | **Baseline** | **30** | **60** | **90** | **120** |
| Do Nothing | 0 | 13.7 | 19.8 | 38.7 | 53.5 | 68.2 |
| Elevate Roads | 9,804,600 | 0.0 | 0.0 | 0.9 | 2.6 | 6.7 |
| Roadside Dike | 8,334,400 | 0.0 | 0.0 | - | - | - |
| Large Dike | 8,429,800 | 0.0 | 0.0 | - | - | - |
| Relocate | 272,540,000 | 0.0 | 0.0 | 0.0 | 0.0 | 0.0 |

Including the airport access road, which begins flooding around 2050, the community AFE increases in the higher sea-level-rise scenarios. The AFE for the entire community is provided in Table 10.

**Table 10.** Estimated costs and AFE for the entire community.

| Measure | Cost (USD 2020) | Estimated AFE, SLR Scenarios (km h/year) | | | | |
|---|---|---|---|---|---|---|
| | | Baseline | 30 | 60 | 90 | 120 |
| Do Nothing | 0 | 13.8 | 20.0 | 40.0 | 57.6 | 79.0 |
| Elevate Roads | 9,804,600 | 0.0 | 0.2 | 2.2 | 6.7 | 13.3 |
| Roadside Dike | 8,334,400 | 0.0 | 0.2 | - | - | - |
| Large Dike | 8,429,800 | 0.0 | 0.2 | - | - | - |
| Relocate | 272,540,000 | 0.0 | 0.0 | 0.0 | 0.0 | 0.0 |

This shows that the roadside dike is the most cost-effective option for preventing road flooding, and that relocation is more expensive than other alternatives by two orders of magnitude and would not make sense in this time interval.

The cost of the road elevation mitigation option was also calculated by multiplying the average AFE between 2020 and 2050 (16.7 km h/year) by the unit cost of flood exposure calculated in goal two (USD27,873/km h) and 30 years. This resulted in an estimated protection cost of USD14,009,700, which is USD4,205,100 more than the estimated road elevation cost calculated based on the compilation of bids.

## 5. Discussion and Conclusions

The Hooper Bay case study investigated the annual road flood exposure with sea level rise, estimated a unit cost of flood exposure, and the costs of potential mitigation measures. Flood exposure was determined by modeling historic storms and re-modeling them with simulated sea level rise, estimating the flood exposure return periods, mapping the flood exposure, and applying the return periods to flooded observation points to calculate the annual flood exposure. Unit cost was found by estimating the access road elevation costs from the Airport Improvements Project, determining the number of years the elevation project was likely to be effective, and calculating the annual flood exposure that would have occurred without the elevation between 2020–2050. An estimate of the annual cost of flooding (for 2020–2050) was calculated by taking the product of the unit cost of flood exposure and the annual flood exposure. Finally, the community road flooding protection costs were calculated by roughly designing three possible mitigation routes (road elevation, putting dikes along the roadways, and creating a large dike that creates polders near the community), estimating the costs of the options based on the Airport Improvements Project, and estimating the resulting AFE.

The extreme value analysis used flood water level to estimate the return period of floods, rather than the kilometer-hours of inundation used later in the study. The kilometer-hours metric is specific to measuring the flood impact on roads and is limited to the observation points chosen for this study—if other infrastructure (for example, a power plant that can never be allowed to flood) is examined, the kilometer-hours of road flooding will be irrelevant to it. By using the maximum water level over the whole community, a more equitable return period is assessed. This may not be the ideal measurement for flooding—floods can cause impact because of duration or current speed, as well as depth. The model area also included ground not essential to the community; storms that caused no flooding in the community might be rated more highly than deserved when the maximum water level is taken from the entire model area (the October 1996 flood, which never touched the community, had a maximum level of 3.06 m). Additional work is needed to determine what metric may align best with flood impact in the whole community.

The large confidence intervals on the extreme value analysis reflect the small sample size and speak to a need for a better method of estimating the return period of flooding. As we have seen, the extent of flooding is not directly linked to storm-surge height. There is a need for additional coastal flood modeling on the Yukon-Kuskokwim Delta to better determine how sea level rise will affect the coastal communities and provide larger sample sizes for this type of analysis.

Mapping the maximum flood water depth had several interesting results. Large stretches of the area experience flood depths within 10 cm of each other, reflecting the very flat ground around Hooper Bay. The maps also appear to corroborate the hypothesis from watching the flood animations that the flooding mostly comes from east of the community. Finally, the map of the flood line shows how the flood waters are likely to rise in the community over time, turning Old Town and New Town into separated islands.

The annual flood exposure analysis demonstrated the success of the airport access road elevation project; not only did it reduce flooding on the airport road to near 0 for the baseline and 30 cm SLR scenarios, the AFE for the entire community was greatly reduced, showing the high impact of the project. With the assumed scenario of 30 cm of SLR in Hooper Bay in 2050, this indicates that the airport access road will begin experiencing storm-surge flooding in just under 30 years.

Comparing the costs of the mitigation measures showed that building a dike may be the best option for the community to mitigate flooding through 2050. The roadside dike is estimated to be the cheapest option. However, it should also be noted that the larger dikes, which cost roughly USD100,000 more than the roadside dike, have the benefit of creating a polder between Old Town and New Town, reclaiming some land from flooding, and possibly enabling community expansion—though further elevation or a pump system in this area may be necessary to make the land fully habitable. The larger dikes also cross stretches of wetlands, which may lead to additional expenses from permitting.

From the cost estimate analysis, it is clear that a large portion of the construction costs are due to set-up and over-head, bringing in construction equipment, sheltering the crew, and other flat costs that incur regardless of project size. This means that to save on costs it may make sense to plan multiple projects to happen simultaneously or sequentially—for example, planning all road elevations to happen at once.

Applying the unit cost of flood exposure to the flood exposure of the entire community resulted in an estimate about USD4.2 million greater than the cost estimate based on the Hooper Bay Airport improvement project. A large part of this is from the set-up and overhead costs, which were assumed to be the same for both elevating the access road and elevating all the roads and amounted to USD3,590,700 (37–45% of total project costs). The community mitigation measure has a lower percentage of these set-up costs in its total project costs than the access road project, and this translates to an overestimate of USD2,698,600 (64% of the difference between the two estimates). The remaining difference may be from poor assumptions of cost drivers in the cost estimation (e.g., that surveying costs scale linearly).

This again demonstrates the possible advantages of doing multiple projects at once; the set-up costs can be minimized. However, this does not show the political advantage that may come from doing smaller, distinct projects over time. Multiple projects simultaneously will have a higher total price tag at the outset compared to smaller projects, and may be easier to acquire funding for, even though they are more expensive over the long term. This is analogous to the decision to relocate, which is very expensive but may absolve all coastal threats.

This research explored a method of evaluating flood risk for coastal Alaska, leveraging flood modeling to determine the flood exposure that the community will face in the coming years. It additionally explored a unit cost of flood exposure, associating flood exposure with its cost of mitigation, and examined three different methods of approaching flood mitigation in Hooper Bay. This showed the flooding that Hooper Bay is likely to experience in the coming years, the possible costs of mitigation, and an apparent advantage for constructing dikes to protect the community compared to elevating all roadways.

This study is non-comprehensive and could greatly benefit from future work. In the Hooper Bay study, future work could improve flood severity return-period analysis. It was shown that flood severity is not directly correlated to storm-surge height because of other factors like storm direction of approach, such that storm surge return periods do not necessarily predict flooding severity. However, a shortage of storm-flooding data points

means that applying extreme value theory directly to floods results in return periods with great uncertainty. Future work could identify factors that, combined with storm surge, predict flood severity, so that those factors could be used to estimate the flood-return period. Alternatively, additional modeling could be done to expand the number of data points, though this may be time and resource intensive.

It was assumed that road-flooding impact can be evaluated through kilometer-hours of flooding, and that the allowable amount of flooding is 0 km hours on the airport road. Using kilometers is an approximate for flood extent, and therefore general community impact. More complicated road networks, where flooding one road has greater impact than another, may need more complex systems analysis to develop a metric. Community communication and surveying are necessary to determine the true allowable annual flood exposure; the community currently experiences 3.2 km hours of flooding per year, but whether this is acceptable is unknown. The study should also be expanded to include additional infrastructure and evaluate erosion and permafrost melt impacts to obtain a more holistic picture of community impact.

Construction cost-estimation could be improved by compiling data from a greater number of projects from the Yukon-Kuskokwim Delta; this could better inform the costs of flood mitigation, and therefore the best strategies for each community. A greater variety of project types should be considered, including utilizing natural features, or combining multiple smaller projects (elevating one section of road, building a short dike somewhere else, etc.). Projects could also be framed for multiple time planning periods by designing to the shorter period and then calculating the cost it would take to improve the project to meet protection levels for the following planning period. Project maintenance costs should also be considered—in this case study, the costs of road and dike upkeep could shift the outcome.

The unit cost of flood exposure could be improved by separating the overhead/lump sum costs from the material costs. The material costs could be approximated through a unitized cost, but lump costs that depend on how a project is split up need to be evaluated separately to avoid over-counting them, as has happened in this study.

The unit cost of flood exposure was originally conceived as a way to estimate the flood risk (cost, USD/y) in multiple communities in a given region. Assuming that the unit cost of flood exposure was reasonably valid for a given region, one could compute the annual flood exposure for each community, and take the product of unit cost of flood exposure and annual flood exposure for each community. This calculation would enable a reasonably objective assessment of the flood risk (USD/y) in multiple communities. Furthermore, if a community-specific unit cost of flood exposure is divided by the population served by the flood mitigation (yielding USD/(kilometer-hours-person) units), a cost-benefit ratio emerges that could be an objective basis for decisions about where to invest in flood mitigation.

**Author Contributions:** Supervision, T.M.R.; Writing—original draft, A.C.M.; Software, A.C.M.; Resources, T.M.R.; Computer modeling, T.M.R. The majority of the hands-on work on the manuscript was undertaken by first author A.C.M., who recently completed her Master's of Science in Civil Engineering Degree. For example, the majority of the computer flood simulations were conducted by A.C.M. Second author T.M.R. conceived of the overall project, provided guidance on the project scope, provided support for the computer modeling, and provided feedback along the way. All authors have read and agreed to the published version of the manuscript.

**Funding:** The work reported here was funded by an NSF grant, with award number 1745508.

**Institutional Review Board Statement:** Not applicable.

**Informed Consent Statement:** Not applicable.

**Data Availability Statement:** Data is available upon request to tmravens@alaska.edu.

**Acknowledgments:** The authors would like to acknowledge the contributions of John Allen and Michael Ulmgren in the development of the Delft3D model.

**Conflicts of Interest:** The authors declare no conflict of interest.

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
