# Peer review of "Assessing Coastal Road Flood Risk in Arctic Alaska, a Case Study from Hooper Bay"

_jmse, doi:10.3390/jmse10030406_

Round 1

Reviewer 1 Report

The manuscript 'Assessing Coastal Road Flood Risk in Arctic Alaska' presents the Hooper Bay Alaska case study related to sea level rise flood mitigation options. The study is based on a number of background data and information, and a variety of analysis. However, the manuscript resembles the first draft of the article and might be published if rewritten and presented in a more concise way.
Please explain the 'use-inspired science' approach mentioned in the Abstract, and find more references that would improve the article. For the start, please check the flood risk definitions in: Tariq, M.A.U.R.; Farooq, R.; van de Giesen, N. A Critical Review of Flood Risk Management and the Selection of Suitable Measures. Appl. Sci. 202010, 8752. https://doi.org/10.3390/app10238752
Than, make a distinction between Flood hazard map (and Flood risk map) and flood exposure.
The figures should be improved and more informative, and some should be merged, like Fig. 10 and 11.
Also, please rethink and pay attention to the differences of the return periods presented in Tab. 5. The statistical analysis presented requires more attention. Please look for the appropriate references that would support the used methods, and check the definition of the annual flood.

Author Response

Reviewer 1:

The manuscript 'Assessing Coastal Road Flood Risk in Arctic Alaska' presents the Hooper Bay Alaska case study related to sea level rise flood mitigation options. The study is based on a number of background data and information, and a variety of analysis. However, the manuscript resembles the first draft of the article and might be published if rewritten and presented in a more concise way.

Please explain the 'use-inspired science' approach mentioned in the Abstract, and find more references that would improve the article. For the start, please check the flood risk definitions in: Tariq, M.A.U.R.; Farooq, R.; van de Giesen, N. A Critical Review of Flood Risk Management and the Selection of Suitable Measures. Appl. Sci. 2020, 10, 8752. https://doi.org/10.3390/app10238752

Author responses:

“Use-inspired science” or “use-inspired research”  is a term coined by the National Research Council (National Research Council, 2008).  Rather than conduct research for the sake of research alone, this type of research is designed to directly support decision making. We have added the NRC reference to the manuscript, and we added the statement: “This study conducts “use-inspired research” (National Research Council 2008) and it has the following goals”..[line 125].

We have reviewed the Tariq et al. 2020 paper with a focus on the flood risk definitions in that paper. How do we define flood risk?  We define risk as the annual cost of flooding ($/y). This is determined as the product of the unit cost of flood exposure [$/km-hr] and annual flood exposure [km-hr/y]. The definition of flood risk employed here is similar to the Tariq et al [2020] definition: risk [$/y] = Probability [y-1] x consequences [$]. However, we have found it expedient to use the annual flood exposure rather than the flood probability. This is because our flood modeling tools allow us to quantify that annual flood exposure. Similarly, we use a unit cost of flood exposure rather than consequences. Again, this is because data available to us allows us to define the unit cost. We have added lines 136-138: “The flood risk calculation we employ resembles the one provided by Tariq et al. 2020: Risk ($/y) = Probability (y-1) x Consequences ($). However, given the model-based flood exposure available to us, we chose to directly tie probability and consequences to flood exposure.”

We have added several references. 

Reviewer 1:

Then, make a distinction between Flood hazard map (and Flood risk map) and flood exposure.

Author responses:

As our above response indicates, we have chosen to base our flood risk assessment based on flood exposure. We believe that we have been clear on this point in the manuscript. We describe our focus on flood exposure particularly in lines: 125-141, but we discuss flood exposure throughout the manuscript (71 times according to word).

Reviewer 1:

The figures should be improved and more informative, and some should be merged, like Fig. 10 and 11.

Author responses:

We have made improvements to the figures in the limited time available. We will make additional improvements if the paper is accepted for publication. In revision 1, we have improved Figure 3 by removing the title. In order to make the paper more concise, we eliminated Figures 10 and 11, but kept the discussion of relocation cost. We improved Figure 10 (formerly Figure 12) by removing the title. 

Reviewer 1:

Also, please rethink and pay attention to the differences of the return periods presented in Tab. 5. The statistical analysis presented requires more attention. Please look for the appropriate references that would support the used methods, and check the definition of the annual flood.

Author responses:

We have reviewed the portion of the manuscript pertaining to the calculation of return periods, which follows the work of Goda (2010). We feel that this is one of the strongest portions of the manuscript, and believe that the work of Goda is authoritative. We point out that our return-period analysis considered multiple approaches including the Gumbel, Weibell, and Freshet distributions. We chose the Weibull distribution as it fit our data best. Our work did not specifically focus on determining the annual flood. Instead, we determined the return period of a number of historic storms, and the road flood exposure (km-hr) of those storms Then, using equation 8, we computed a weighted average of the storm road flood exposures to determine the Annual Flood Exposure (AFE), which is the expected amount of road flood exposure in a given year. 

Reviewer 2 Report

This review is about the article “Assessing Coastal Road Flood Risk in Arctic Alaska”; by Anna Christina Miller and Thomas Michael Ravens. The article focuses on Assessing Coastal Road Flood Risk in Arctic Alaska covering the Hooper Bay area in Alaska as a case study. It demonstrates a method of estimating road flood risk in coastal Alaskan communities using the community of Hooper Bay Alaska. The study evaluates the current and future flood exposure of roads in Hooper Bay Alaska, proposes a unit cost of flood exposure to estimate the cost of flooding, and compares various mitigation efforts including elevating roads and building dikes.

In general, I found the manuscript very informative, clear, well written, and easy to follow. It should be considered for publication. However, the following points/comments/suggestions may be incorporated before publishing.

Line 2: Article Title

This Manuscript is about a case study conducted in the Hooper Bay area in Alaska and it will not cover the entire area of the Arctic region of Alaska but a small area of it. Henceforth, I believe the current title is misleading and should be realigned to its context.

Therefore, I suggest changing the title to; Assessing Coastal Road Flood Risk in Arctic Alaska. A Case Study from Hooper Bay OR any appropriate change to the title.

Line 06-20: Abstract

The current Abstract has mostly discussed the outline of the aims and objectives, methodology (somewhat similar to repeating the methodology section), and two general statements about analysis. Perhaps authors can rethink reducing/concise the methodology statements in the abstract and add more specific results obtained in this study. Ex: overview about unit cost of flood exposure, Average annual flood exposure (km-hrs/year), … etc as the case study findings. It would be also better to include a statement or two about the authors' perspectives on how to transform results obtained in the case study to a broader area (Arctic Alaska) to map the main title. It is recommended to add a paragraph about the same to section 5. Discussion and Conclusions (Line 542-656).

Line 21: Keywords

The given keywords are too general. Better to have a few more keywords covering the subject area or a highly focused specific area covering the scope of the manuscript.

Line 25-28, 51-53: Citation missing. One major weak point noticed in this manuscript is that some of the statements throughout the article are not properly supported by respective citation(s). Also, it was noticed a few articles were referred, henceforth it is recommended to enhance the reference list (if possible) with the recent articles.

Line 77-78: Please add geographical coordinates into the statement: ex: 61°31′44″N 166°5′46″W (61.528980, -166.096196)

Line 85-86: Please add Hooper bay geographical location and map dimensions, and if possible add a sub-map to illustrate Arctic Alaska. In the Figure 1 caption; Location and Map of Hooper Bay: what do you mean by Location? Better to expand the caption to have a clear reflection about what do u meant by location.

Line 157-158: Figure 2 should be replaced by a high-resolution image. Map dimensions should be inserted including in the few other figures. In some figures scales in CGS units (ex: Figure 6) and some places, it is in SI units

Line 218-219: Better to indicate the R2 value. Better to remove the title of the graph and add the information into axis labels. Look at the other figures for similar changes.

Line 439-442: In Figures 10 and 11, better to have the same range for X-axis (population) to increase the readability; ie from 300 to 1500 or as appropriate.

Author Response

Reviewer 2:

This review is about the article “Assessing Coastal Road Flood Risk in Arctic Alaska”; by Anna Christina Miller and Thomas Michael Ravens. The article focuses on Assessing Coastal Road Flood Risk in Arctic Alaska covering the Hooper Bay area in Alaska as a case study. It demonstrates a method of estimating road flood risk in coastal Alaskan communities using the community of Hooper Bay Alaska. The study evaluates the current and future flood exposure of roads in Hooper Bay Alaska, proposes a unit cost of flood exposure to estimate the cost of flooding, and compares various mitigation efforts including elevating roads and building dikes.

In general, I found the manuscript very informative, clear, well written, and easy to follow. It should be considered for publication. However, the following points/comments/suggestions may be incorporated before publishing.

Line 2: Article Title

This Manuscript is about a case study conducted in the Hooper Bay area in Alaska and it will not cover the entire area of the Arctic region of Alaska but a small area of it. Henceforth, I believe the current title is misleading and should be realigned to its context.

Therefore, I suggest changing the title to; Assessing Coastal Road Flood Risk in Arctic Alaska. A Case Study from Hooper Bay OR any appropriate change to the title.

Author response:

We agree with the suggestion of reviewer 2, and we have changed the title accordingly.

Reviewer 2:

Line 06-20: Abstract

The current Abstract has mostly discussed the outline of the aims and objectives, methodology (somewhat similar to repeating the methodology section), and two general statements about analysis. Perhaps authors can rethink reducing/concise the methodology statements in the abstract and add more specific results obtained in this study. Ex: overview about unit cost of flood exposure, Average annual flood exposure (km-hrs/year), … etc as the case study findings. It would be also better to include a statement or two about the authors' perspectives on how to transform results obtained in the case study to a broader area (Arctic Alaska) to map the main title. It is recommended to add a paragraph about the same to section 5. Discussion and Conclusions (Line 542-656).

Author response:

We appreciate the spirit of the comment. In order to make the abstract more concise, one could remove the two sentences: “Nine historic storms and their associated flood depths were subject to return period analysis and modeled for several sea level rise scenarios. Based on the simulated road flood exposure (km-hours/storm), and the storm return period, an annual flood exposure (km-hours/year) was computed.” However, we prefer to leave these two sentences in. We believe the methodology developed in the paper is distinctive, and we would like to include some mention of the methodology in the abstract if possible. We have added two sentences at the end of the abstract: “The flood risk calculation, based on the unit cost of flood exposure, could be made for all of the communities in a given region to identify those communities that face a high flood risk. Further, if one divides the unit cost by the population, one obtains a cost/benefit ratio that potentially could be used to prioritize flood mitigation work.” 

Finally, based on the recommendation of reviewer 2, we have added the following paragraph to the Discussion and Conclusions section: “The unit cost of flood exposure was originally conceived as a way to estimate the flood risk (cost, $/yr) in multiple communities in a given region. Assuming that the unit cost of flood exposure was reasonably valid for a given region, one could compute the annual flood exposure for each community, and take the product of unit cost of flood exposure and annual flood exposure for each community. This calculation would enable a reasonably objective assessment of the flood risk ($/y) in multiple communities. Further, if a community-specific unit cost of flood exposure is divided by the population served by the flood mitigation (yielding $/(km-hours-person) units), a cost-benefit ratio emerges that could be an objective basis for decisions about where to invest in flood mitigation.”

Reviewer 2:

Line 21: Keywords

The given keywords are too general. Better to have a few more keywords covering the subject area or a highly focused specific area covering the scope of the manuscript.

Author response:

We have made the key words more specific. They now read: “Alaska coastal flooding; Alaska flood risk estimation and mitigation”

Reviewer 2:

Line 25-28, 51-53: Citation missing. One major weak point noticed in this manuscript is that some of the statements throughout the article are not properly supported by respective citation(s). Also, it was noticed a few articles were referred, henceforth it is recommended to enhance the reference list (if possible) with the recent articles.

Author response:

We have added several new references.

Reviewer 2:

Line 77-78: Please add geographical coordinates into the statement: ex: 61°31′44″N 166°5′46″W (61.528980, -166.096196)

Author response:

We have added the requested geographic coordinates.

Reviewer 2:

Line 85-86: Please add Hooper bay geographical location and map dimensions, and if possible add a sub-map to illustrate Arctic Alaska. In the Figure 1 caption; Location and Map of Hooper Bay: what do you mean by Location? Better to expand the caption to have a clear reflection about what you meant by location.

Author response:

We have added the Hooper Bay geographical location and map dimensions in Figure 1. The inset map actually provides the location of the US Arctic. We have added the sentence:  “The YK Delta lies on the Bering Sea, and this Sea marks the southern portion of the US Arctic (Arctic Council, 2022).” [lines 90-91]

Reviewer 2:

Line 157-158: Figure 2 should be replaced by a high-resolution image. Map dimensions should be inserted including in the few other figures. In some figures scales in CGS units (ex: Figure 6) and some places, it is in SI units.

Author response:

We have replaced Figure 2 with a high-resolution image. The map dimensions (in SI units) have been added to the other figures. 

Reviewer 2:

Line 218-219: Better to indicate the R2 value. Better to remove the title of the graph and add the information into axis labels. Look at the other figures for similar changes.

Author response:

We have made the recommended changes. 

Reviewer 2:

Line 439-442: In Figures 10 and 11, better to have the same range for X-axis (population) to increase the readability; ie from 300 to 1500 or as appropriate.

Author response: 

For brevity, we decided to remove Figures 10 and 11, and to include the information in the text.

Round 2

Reviewer 1 Report

The manuscript 'Assessing Coastal Road Flood Risk in Arctic Alaska, a Case Study from Hooper Bay' is improved compared to the previous version. 

The present article type is 'Article'. One of the requests for this article type is: 'The article should include the most recent and relevant references in the field.' Still, the literature review is a weak point of the manuscript.

lines 197-199 'a particular extremity will occur each year'. According to the probability definition, a particular extremity will be exceeded at least once in the return period.

line 516 - title 'Flood maps'. Please be more specific both in the title and the text. There are several types of flood maps.

Author Response

Second Round, Reviewer 1:

Reviewer 1:

The manuscript 'Assessing Coastal Road Flood Risk in Arctic Alaska, a Case Study from Hooper Bay' is improved compared to the previous version. 

The present article type is 'Article'. One of the requests for this article type is: 'The article should include the most recent and relevant references in the field.' Still, the literature review is a weak point of the manuscript.

Authors:

We have added additional references and discussion in lines 61-69 of the revised manuscript.

Reviewer 1:

lines 197-199 'a particular extremity will occur each year'. According to the probability definition, a particular extremity will be exceeded at least once in the return period.

Authors:

We have revised the statement to read: “The return period provides the likelihood that a flood of a particular extremity will occur in a given year; for example, a 100-year flood would have a 1/100 chance in a given year. Alternatively, the return period can be considered the recurrence interval, which is the expected average time between storms of that magnitude. It is expected that at least one storm of that magnitude occurs within the return period.”

Reviewer 1:

line 516 - title 'Flood maps'. Please be more specific both in the title and the text. There are several types of flood maps.

Authors:

We have changed the section title to “Map of maximum flood extent and depth.” The text within that section has also been changed accordingly.